# Chemical Reaction Networks Possess Intrinsic, Temperature-Dependent Functionality

**DOI:** 10.3390/e22010117

**Published:** 2020-01-18

**Authors:** Stephan O. Adler, Edda Klipp

**Affiliations:** Theoretical Biophysics, Humboldt-Universität zu Berlin, 10099 Berlin, Germany; adlerste@hu-berlin.de

**Keywords:** temperature dependence, flux reversal, entropy production density

## Abstract

Temperature influences the life of many organisms in various ways. A great number of organisms live under conditions where their ability to adapt to changes in temperature can be vital and largely determines their fitness. Understanding the mechanisms and principles underlying this ability to adapt can be of great advantage, for example, to improve growth conditions for crops and increase their yield. In times of imminent, increasing climate change, this becomes even more important in order to find strategies and help crops cope with these fundamental changes. There is intense research in the field of acclimation that comprises fluctuations of various environmental conditions, but most acclimation research focuses on regulatory effects and the observation of gene expression changes within the examined organism. As thermodynamic effects are a direct consequence of temperature changes, these should necessarily be considered in this field of research but are often neglected. Additionally, compensated effects might be missed even though they are equally important for the organism, since they do not cause observable changes, but rather counteract them. In this work, using a systems biology approach, we demonstrate that even simple network motifs can exhibit temperature-dependent functional features resulting from the interplay of network structure and the distribution of activation energies over the involved reactions. The demonstrated functional features are (i) the reversal of fluxes within a linear pathway, (ii) a thermo-selective branched pathway with different flux modes and (iii) the increased flux towards carbohydrates in a minimal Calvin cycle that was designed to demonstrate temperature compensation within reaction networks. Comparing a system’s response to either temperature changes or changes in enzyme activity we also dissect the influence of thermodynamic changes versus genetic regulation. By this, we expand the scope of thermodynamic modelling of biochemical processes by addressing further possibilities and effects, following established mathematical descriptions of biophysical properties.

## 1. Introduction

Living beings are subject to fluctuations of environmental factors they need to adapt to. This process of adaptation is called acclimation and can happen on different levels of organization. There can be morphological changes [1], but also a reorganization of biochemical processes like metabolism [2]. Many plants, for example, accumulate carbohydrates when their growth temperature is low, which is assumed to increase freezing tolerance by serving as osmoprotection [3,4]. As plants cannot flee from cold or heat, it is especially important for them to be able to adapt to such fluctuations.

The changes in biological processes differ in their nature and cause. Some are known to be triggered responses like the production of heat shock proteins as a consequence of thermal or other stresses [5]. These responses are coordinated by regulatory mechanisms involving, i.e., gene expression alteration or post-translational modifications that actively promote the desired biochemical changes.

Furthermore, there are thermodynamic effects that directly influence the involved biochemical reactions themselves. As reaction rates depend on temperature, different reactions can change their velocities to varying amounts leading to a rearrangement of flux distributions within the biochemical systems. Additionally, enzyme activity and protein stability change [6,7], contributing to the described effects. The observed responses can be unexpected or even contradictory. For example, most enzyme activities increase with temperature up to a thermal optimum [6], nevertheless, an overall increase in activity of Calvin cycle enzymes has been reported for *Arabidopsis thaliana* during cold treatment [8]. A review on changes in aerobic metabolism of ectotherms is provided by Schulte (2015) [9].

Sometimes it is not straightforward to determine whether an observed change is part of the acclimation process or merely a thermodynamic byproduct. At the same time, observations might be attributed to some unknown regulation mechanism although they are caused by the thermodynamically induced changes within the reaction network. Additionally, parts of the acclimation process are hard to identify as they compensate for changes that would occur otherwise. In these cases, the crucial observation would be the absence of a change.

Ruoff et al. (2007) [10] used a systems biology approach to demonstrate how reaction networks can possess the inherent property to compensate for temperature changes as a result of the relation between its architecture and the properties of the involved reactants and enzymes that determine the activation energies and equilibrium constants of the corresponding reactions [10]. This type of compensation does not need any active regulation and can be referred to as static compensation.

By introducing control coefficients, they also showed and quantified how changes to single reactions can propagate and shape the overall behavior of whole reaction schemes. These changes can, for example, be enzyme modifications affecting the value of apparent activation energies and therefore changing the balance of the reaction network. As this type of compensation involves changes of network properties, it can be referred to as dynamic compensation. However, temperature changes can thermodynamically affect biochemical reaction networks in different ways and the network responses can be of different nature than compensation.

In this study, we apply the same systems biology approach to simple network motifs and demonstrate cases of temperature induced reversal of fluxes and how temperature changes can regulate and redirect fluxes in branched reaction systems. We explicitly ignore all other potential changes in gene expression, or at other regulatory levels the organism could employ to prevent or enhance the effects of varying temperature. Thus, we only study the passive rebalancing of fluxes that can ultimately lead to the accumulation of certain reactants without any active mechanistic regulation on a molecular level. These results therefore represent inherent functional network features and contribute to addressing the question to what extent observed changes in network behavior, or their absence, are the consequence of active regulation and what the role is of intrinsic network properties in this context.

## 2. Materials and Methods 

In order to analyze the effects of temperature changes on different reaction networks, we chose three common network motifs with qualitatively different structures: a linear pathway, a branched pathway and a circular network motif. For each of the three networks, we defined a system of ordinary differential equations (ODEs) describing their dynamics.

The linear pathway:(1)$$\frac{d}{dt}A=k_{+1}-k_{-1}A-k_{+2}A+k_{-2}B\;\frac{d}{dt}B=k_{+2}A-k_{-2}B-k_{+3}B+k_{-3}$$

The branched pathway:(2)$$\frac{d}{dt}A=k_{+1}-k_{-1}A-k_{+2}A-k_{+3}A+k_{-2}B+k_{-3}C\;\frac{d}{dt}B=k_{+2}A-k_{-2}B-k_{+4}B+k_{-4}\;\frac{d}{dt}C=k_{+3}A-k_{-3}C-k_{+5}C+k_{-5}$$

The circular pathway:(3)$$\frac{d}{dt}A=k_{3}C-k_{4}A-k_{6}A\;\frac{d}{dt}B=k_{1}+k_{4}A-k_{2}B\;\frac{d}{dt}C=k_{2}B-k_{3}C-k_{5}C$$

To describe the effect of temperature on each reaction, we assigned the famous Arrhenius equation [11] that holds for most chemical reactions within a temperature range, to each of the rate constants. It describes how the rate constant *k* of a chemical reaction depends on temperature *T* and the Gibbs energy of activation *ΔG^‡^*.
(4)$$k=A\cdot \exp\left(-\frac{\Delta G^{‡}}{RT}\right)$$

In Equation (4), *R* is the universal gas constant and *A* is the so-called pre-exponential factor. This factor describes the frequency of molecule collisions. It is specific for each reaction and depends on the molecular structures of the reactants. There exists different approaches to determine and describe *A*. These are either empirical [11], based on collision theory [12] or transition state theory [13], with varying focus and complexity. Technically, *A* depends on temperature as well, but the magnitude of this dependency is comparably small to that of the exponential expression and can therefore be neglected in this study.

This description of temperature dependence of chemical reaction rates can be applied to mass-action kinetics, but also to Michaelis-Menten kinetics that describe enzymatic reactions, as explained in Bozlee (2007) [14] for the irreversible case. Applying these principles to reversible Michaelis-Menten kinetics
(5)$$v=\frac{\frac{v_{max}^{f}S}{K_{M}^{S}}-\frac{v_{max}^{b}P}{K_{M}^{P}}}{1\;+\;\frac{S}{K_{M}^{S}}\;+\;\frac{P}{K_{M}^{P}}}$$
results in
(6)$$v_{max}^{f}=k_{+2}{\left[E\right]}_{tot}=A_{+2}\exp\left(\frac{G_{4}-G_{3}}{RT}\right){\left[E\right]}_{tot}$$
(7)$$v_{max}^{b}=k_{-1}{\left[E\right]}_{tot}=A_{-1}\exp\left(\frac{G_{2}-G_{3}}{RT}\right){\left[E\right]}_{tot}$$
with
(8)$${\left[E\right]}_{tot}=\left[E\right]+\left[ES\right]$$
and
(9)$$K_{M}^{S}=\frac{k_{-1}+k_{+2}}{k_{+1}}=\exp\left(\frac{G_{3}}{RT}\right)\left(\frac{1}{A_{1}}+\frac{A_{+2}}{A_{+1}}\exp\left(\frac{G_{2}-G_{4}}{RT}\right)\right)$$
(10)$$K_{M}^{P}=\frac{k_{-1}+k_{+2}}{k_{-2}}=\exp\left(\frac{G_{3}-G_{5}}{RT}\right)\left(A_{2}+\frac{A_{-1}}{A_{-2}}\exp\left(\frac{G_{4}-G_{2}}{RT}\right)\right)$$
for the maximum rates $v_{\max}^{f}$ and $v_{max}^{b}$ of the forward and backward reaction, respectively, and the Michaelis constants $K_{M}^{S}$ and $K_{M}^{P}$ of substrate and product, respectively. Here, *G*_1_ to *G*_5_ represent the different levels of Gibbs free energy for each of the reaction steps as illustrated in Figure 1. *A_i_* is the fraction of pre-exponential factors of the forward and backward reaction $\frac{A_{+i}}{A_{-i}}$.

To find parameter sets that result in particular temperature responses, we started to systematically vary Gibbs energies of activation (*ΔG^‡^*) and pre-exponential factors (*A*) within the networks one-by-one and collectively and simulated the systems at different temperatures from 273 to 308 K until they reached steady state. The temperature interval was chosen to represent a broad but physiological range for most organisms. After observing the system responses, we were able to manually adjust the parameters for sets that turned out promising to result in a certain behavior. In this process, all parameters were chosen to be within a reasonable range in the order of magnitude comparable to the ones provided by Ruoff et al. (2007) [10] and Bozlee (2007) [14]. The used parameters for the presented simulations are provided in Table 1.

All presented networks were implemented using the programming language Python and the corresponding ODE systems were solved using the ‘odeint’-function of scipys integrate package that utilizes the LSODA algorithm from the FORTRAN library odepack to compute a numerical solution.

## 3. Results

Since most observations regarding metabolic changes caused by temperature shift concern long term changes, it is reasonable and necessary to examine non-transient responses and focus on steady state fluxes. Temperature rises and declines seasonally or during the course of a day, causing long-term responses, while metabolic fluxes adapt within seconds leading to strong time scale separation. Most fluxes simply increase with higher temperature, resulting in an exponentially increasing response curve as shown in Figure 2A.

Since biological networks most often contain some reversibility and branching, the effective flux-temperature response curves can have very different shapes, as already demonstrated for CO_2_ fixation in Ruoff et al. (2007) [10]. Interestingly, besides flux compensation, this might result in temperature-dependent functional network features like flux reversal or pathway selection.

### 3.1. The Direction of Fluxes Can be Reversed by Solely Changing Temperature

A major change to fluxes that could drastically alter the response of a network would be if fluxes changed their direction. For this to happen, the unidirectional flux at low temperatures in one direction (e.g., forward) must be at a higher level than the one in the opposite direction (then backward), but the corresponding temperature response curve should be relatively flat over a broad temperature range. If the temperature response curve of the unidirectional flux in the opposite direction (backward) is steeper and close enough to the one in forward direction, the net flux is zero at the temperature where the curves intersect. Above this temperature, the flux is reversed as illustrated in Figure 2B.

Since the molecules are converted into other molecules with different shapes and physical properties, the pre-exponential factors and Gibbs energies of activation for the forward and backward direction of a reaction can differ greatly; so does the ratio of these parameters that defines the shapes of the resulting flux temperature response curves, which makes the described scenario plausible.

This effect does not even demand for a complex network structure, since it is simply based on the physical properties of the involved reactants, but could potentially reverse the flux direction of a complete linear pathway as illustrated in Figure 3A.

If we look at a system of reversible reactions arranged in a linear sequence and consisting of *i* species *S_i_*, a precursor *P*_1_ and a product *P*_2_ (Figure 4B), we can describe each reaction rate using mass action kinetics as
(11)$$v_{i}=k_{+i}S_{i-1}-k_{-i}S_{i}.$$

For each of the reversible reactions *i*, we can write the equilibrium constant *K_i_* as
(12)$$K_{i}=\frac{k_{+i}}{k_{-i}}=\frac{A_{+i}\exp\left(-\frac{\Delta G_{+i}}{RT}\right)}{A_{-i}\exp\left(-\frac{\Delta G_{-i}}{RT}\right)}=A_{i}\exp\left(\frac{\Delta G_{-i}-\Delta G_{+i}}{RT}\right).$$

From this and considering the Arrhenius law, we can find an expression for the temperature- dependent steady state concentration of each species in the system:(13)$$\begin{array}{c} S_{i}\left(T\right)=\frac{k_{+i}\left(T\right)S_{i-1}-k_{-\left(i+1\right)}\left(T\right)S_{i+1}}{k_{-i}\left(T\right)+k_{+\left(i+1\right)}\left(T\right)}\\ \;\;\;\;\;\;=\frac{A_{+i}\exp\left(\frac{-\Delta G_{+i}}{RT}\right)S_{i-1}+A_{-\left(i+1\right)}\exp\left(\frac{-\Delta G_{-\left(i+1\right)}}{RT}\right)S_{i+1}}{A_{-i}\exp\left(\frac{-\Delta G_{-i}}{RT}\right)+A_{+\left(i+1\right)}\exp\left(\frac{-\Delta G_{+\left(i+1\right)}}{RT}\right).} \end{array}$$

Assuming the concentrations of *P_1_* and *P_2_* as constant, we can also derive a formula for the temperature-dependent steady state flux through the system:(14)$$J\left(T\right)=\frac{P_{1}\prod_{j=1}^{r}K_{j}\left(T\right)-P_{2}}{\sum_{j=1}^{r}\frac{1}{k_{+j}}\prod_{m=j}^{r}K_{m}\left(T\right)}=\frac{P_{1}\prod_{j=1}^{r}A_{j}\exp\left(\frac{\Delta G_{-j}-\Delta G_{+j}}{RT}\right)-P_{2}}{\sum_{j=1}^{r}\frac{1}{A_{+j}}\exp\left(\frac{\Delta G_{+j}}{RT}\right)\prod_{m=j}^{r}A_{m}\exp\left(\frac{\Delta G_{-m}-\Delta G_{+m}}{RT}\right)}$$
with the number of reactions r, the equilibrium constants *K_i_* and the Gibbs free energies of activation *ΔG_+i_* and *ΔG_−i_* for the forward and backward direction of reaction *i*, as illustrated in Figure 4A.

When looking at the numerator, we see that this term can equate to zero and change signs. The temperature at which the flux is zero is the critical temperature (*T_J_*_=0_) at which the overall flux changes direction when it is passed. For *J* = 0 and *ΔG_j_* = *ΔG_−j_* − *ΔG_+j_*, it holds.
(15)$$\prod_{j=1}^{r}A_{j}\exp\left(\frac{\Delta G_{j}}{RT}\right)=\frac{P_{2}}{P_{1}}.$$

If we assume a system of reactions where the values of all pre-exponential factors and Gibbs free energies are relatively close to each other, we can approximate *A_j_* ≈ *A_j+_*_1_ = *A* and *ΔG_j_* ≈ *ΔG_j+_*_1_ = *ΔG*, resulting in the expression
(16)$$A^{r}\exp\left(\frac{r\Delta G}{RT}\right)=\frac{P_{2}}{P_{1}}.$$

We can now finally determine *T_J_*_=0_ for the linear pathway as
(17)$$T_{J=0}=\frac{r\Delta G}{R}{\left(\ln\left(\frac{P_{2}}{P_{1}A^{r}}\right)\right)}^{-1}$$

When looking at single reaction *i* in any pathway, we consider the concentrations of the corresponding substrates and products *S_i-_*_1_ and *S_i_*, the change of Gibbs free energy *ΔG_i_* and the ratio of pre-exponential factors *A_i_*. The critical temperature for flux reversal can be determined as
(18)$$T_{J=0,i}=\frac{\Delta G_{i}}{R}{\left(\ln\left(\frac{S_{i-1}}{S_{i}A_{i}}\right)\right)}^{-1}.$$

In the case of Reaction 2 of the linear network shown in Figure 3, this equates to about 301 K, the temperature at which the pathway indeed changes its flux direction. To further elaborate this, we performed simulations with periodically changing temperature as shown in Figure 3B. Here, we simulated the behavior of the linear pathway in response to a cyclically changing temperature over time for three different temperature regimes. One of the temperature curves cycles around the critical temperature *T_J=_*_0_ at about 301 K. Another temperature curve is set to lower temperatures around 290 K and the last one serves higher temperatures around 310 K. As expected, the resulting flux response curve for temperatures around 300 K changes its sign periodically, switching from forward to backward direction. In contrast to this, the flux at higher temperatures keeps forward direction, only varying in extent. The same holds for the flux at lower temperatures; only here the maintained flux direction is backwards. Interestingly, there are big differences in the amplitudes of these oscillations, increasing from lower temperatures with small amplitudes to higher temperatures with bigger amplitudes, although the absolute changes in temperature for all three temperature curves are the same. Additionally, we notice that for positive (forward) fluxes, the concentration of species B is always higher than the concentration of A and vice versa for negative (backward) fluxes. Also, the concentrations of both species are generally lower for higher temperatures.

### 3.2. Temperature Can Potentially Direct Fluxes Through Selective Branched Systems

Another major alteration of the overall behavior of a network that also involves the network structure would be the temperature-dependent redirection of fluxes to different network branches. We designed a simple branched network and tested different distributions of activation energies and pre-exponential factors for all corresponding reactions and found constellations that resulted in exactly this behavior, as illustrated in Figure 5.

Looking at the lower temperature region, marked in blue, the fluxes enter from the left and upper branch, forming species A and B and leave through the lower branch via C. The intermediate temperature range, marked in red, is characterized by flux separation. While two incoming fluxes result in a single outgoing flux within the blue temperature region, here a single flux is split into two outgoing fluxes. It enters from the left branch forming A which is converted to both, B and C. The separated flux then leaves the network through the upper and lower branch. Eventually, there is a high temperature range, marked as green. Here the network behaves the same as in the blue region, but the roles of the upper and lower branches are reversed. So the fluxes enter through the left and lower branch, forming species A and C and leave through the upper branch via B. In this context, it is important to note that the transitions from one flux mode to another occur at temperatures where one or more of the reversible reactions change their direction of flux, satisfying Equation (18), as described for single reactions and linear pathways in 3.1. At the transition from the blue to the red flux mode, reactions 2 and 4 (upper branch) are reversed, followed by the reversal of reactions 3 and 5 (lower branch) at the transition from the red to the green regime.

### 3.3. Dissecting Influence of Temperature Change Versus Genetic Regulation Resulting in Change of Enzyme Activity

As already pointed out, changes in biochemical processes upon temperature change can be passive consequences resulting from the physical properties of the involved substances and their surroundings, but are also actively regulated on the gene expression and post-translational level. In order to compare and dissect the magnitudes of influence of these different types of response, we designed a minimal branched network, following the same principles as before, and systematically altered temperature and the concentration of one of the involved enzymes (enzyme 2) as illustrated in Figure 6. 

As a measure for the contribution of either enzyme changes or temperature change, we were interested in a measure that is independent of the units of these two modifiers. Therefore, we compared the concentration of substrate *S* upon alteration of both modifiers (*T* and enzyme 2, lower left panel) to the concentration of *S*, when only changing the concentration of enzyme 2 (lower middle panel) or the temperature (lower right panel).

We see that the magnitude of the influence of enzyme concentration changes is bigger than sole temperature changes since fluxes and concentrations both change more drastically in response to it (note the different scales). Nevertheless, when comparing the shapes of the flux response curves, we also see that the type of response differs qualitatively, and when looking at the combined effects on the concentration of *S*, we find that temperature modulates the response to enzyme changes as it shifts the general abundance of *S*.

### 3.4. A Minimal Calvin Cycle Can Exhibit an Increased Flux Towards Carbohydrates at Low Temperatures

We also further analyzed the potential of the minimal Calvin–Benson cycle network presented by Ruoff et al. (2007) [10]. It is a minimal representation of the important cyclic reaction scheme also referred to as dark reaction of photosynthesis. The corresponding reactions comprise the essential processes of carbon fixation from CO_2_ by Rubisco, the reduction of the fixation product and the regeneration of ribulose-1,5-bisphosphate for the next fixation. When taking a closer look at the structure of this network, one realizes that it’s branching provides the potential to show bell-shaped temperature response curves, similar to the ones presented for CO_2_ fixation, at the other branch points as well. Consequently, we were able to identify a parameter set for which the network exhibits an increased flux towards carbohydrates at low temperatures, as frequently reported in literature [4,8]. The network and the corresponding temperature response curve are illustrated in Figure 7.

### 3.5. Connection to Entropy Production

We are interested in the effect that temperature *T* has on the magnitude and the sign of the velocity of biochemical reactions in a network of reactions. In equilibrium, the direction of an isolated reaction *r* is given by the difference in the Gibbs free energy, $\Delta G_{r}\left(S,P\right)$, of its substrate *S* and its product *P*. We can, instead, also use the affinity
(19)$$A_{r}=-\sum_{i=1}^{K}\mu_{i}n_{ir}$$
which represents the (negative) sum over the chemical potentials $\mu_{i}=\mu_{i}{}^{0}\left(p,T\right)+RT\ln c_{i}$ times the stoichiometric coefficients with which they enter the reactions. The affinity has the same value as the difference in Gibbs free energy, when considering the conversion of 1 mol of substrate, but the opposite sign. Hence, a reaction is favorable and it will proceed with a positive rate, when the affinity is positive. 

An important characterization of the system is the entropy production density σ [15,16], which can be expressed as follows
(20)$$\sigma ={\vec{J}}_{Q}\mathrm{grad}\left(\frac{1}{T}\right)-\sum_{i=1}^{K}{\vec{J}}_{c_{i}}\mathrm{grad}\left(\frac{\mu_{{}_{i}}}{T}\right)+\sum_{r=1}^{R}\frac{v_{r}A_{r}}{T}$$

σ is the sum of the product of generalized forces and generalized fluxes. The relevant forces considered for biochemical reaction systems are the temperature gradient $\mathrm{grad}\left(\frac{1}{T}\right)$, the gradient of the chemical potential $\mathrm{grad}\left(\frac{\mu_{i}}{T}\right)$ and the reaction affinity $\frac{A_{r}}{T}$. The respective fluxes are the heat flow ${\vec{J}}_{Q}$, the diffusion flows ${\vec{J}}_{c_{i}}$ and the reaction velocities $v_{r}$. For the specific conditions investigated in this study, we assume that we have a well-mixed system with no spatial gradients, neither of the temperature nor of the chemical potentials. Hence, the first two terms in the expression for the entropy production density vanish and we obtain
(21)$$\sigma =\sum_{r=1}^{R}\frac{v_{r}A_{r}}{T}$$

To link this expression to the considerations on the Arrhenius equation, we assume for simplicity that all reaction rates obey mass action kinetics. Hence, the rate for the forward reactions and backward reactions read, respectively:(22)$$v_{f,r}=k_{f,r}\prod_{i=1}^{K}c_{i,f}^{-n_{i,f}};\;v_{b,r}=k_{b,r}\prod_{i=1}^{K}c_{i,b}^{n_{i,b}}$$
and the net rate is given by the difference of the forward and backward rates.
(23)$$v_{r}=v_{f,r}-v_{b,r}=v_{f,r}\left(1-\frac{\prod_{i=1}^{K}c_{i}^{n_{i,r}}}{K_{eq,r}}\right)$$
Here, $K_{eq,r}=\prod_{i=1}^{K}{\left(c_{i}^{n_{i,r}}\right)}_{eq}$ is the equilibrium constant of the *r*-th reaction.

The affinity can also be expressed as function of reactant concentrations and equilibrium constant, as follows:(24)$$A_{r}=-\sum_{i=1}^{K}\mu_{i}n_{ir}=-RT\ln\left(\frac{\prod_{i=1}^{K}c_{i}^{n_{i,r}}}{K_{eq,r}}\right)$$

Combining the expression for the rates and for the affinity, we easily derive the following expression for the entropy production density:(25)$$\sigma =\sum_{r=1}^{R}\frac{v_{r}A_{r}}{T}=-\sum_{r=1}^{R}v_{r}R\ln\left(1-\frac{v_{r}}{v_{r,f}}\right)$$

It is important to note that this expression is not explicitly dependent on the temperature *T*. However, it is implicitly dependent on *T* via the Arrhenius relations.

Applying this expression to the linear reaction network, shown in Figure 3, results in the temperature response curve for *σ* shown in Figure 8, considering only internal fluxes. We see that, here, entropy production has a clear minimum where it becomes zero. This minimum relates to the temperature at which also the net flux is zero, i.e., *T_J_*_=0_. For temperatures away from this balanced state, *σ* increases in both directions, showing that it does not necessarily decrease with *T* as previous terms might suggest.

## 4. Discussion

In this work, we demonstrate how the interplay of thermodynamic properties and the structure of reaction networks can potentially result in surprising temperature-dependent effects. These effects can be the compensation of flux changes due to temperature shifts, as already demonstrated by Ruoff et al. (2007) [10], but also functional network properties like the reversal of fluxes or the temperature sensitive flux redirection within branched pathways, as shown in this work. Furthermore, we find clues indicating that some of the changes observed in plant metabolism after temperature changes might just occur naturally, at least to some extent, without additional regulation. 

The Arrhenius equation was derived from experimental studies [11] and there are limits to its applicability. The temperature-dependent kinetics utilized in this study are not universal for biochemical reaction networks. The actual physics behind many of such reactions might be more complex than considered here, rendering temperature-dependent modelling very complicated and hard to tackle. Especially because of this, it is even more useful and necessary to not overcomplicate things from the beginning. It has turned out that the Arrhenius equation is actually able to accurately describe the temperature dependency of many reaction rates and, therefore, represents a proven and sufficiently simple instrument to begin with.

Following this approach, we were able to demonstrate the potential for significant changes in the overall network behavior, without any mechanistic regulation, by solely varying temperature, even for the small and elementary network motifs presented in this study. The number of noteworthy effects, within the framework of the approach, might be much higher, considering the multitude of combinations of network structures, reaction kinetics and parameter distributions that are possible, and it remains a challenge to fully explore the space of possibilities at this point. For future investigations, it would be interesting to study the role of different kinetics in this context, by utilizing temperature-dependent, reversible Michaelis-Menten kinetics as explained in the Methods section. In this way, one can describe enzymatic reactions, also considering the role of changing enzyme concentration, as this can be a limiting factor in a cellular environment. However, to approach the problem in a simple and structured manner, we focused on non-complex effects that are relatively easy to assess and comprehend.

The reversal of reaction fluxes falls exactly into this category. It can fundamentally change the overall behavior of a biochemical reaction network and can be observed in vivo. It has been shown by Maitra and Lobo (1978) [17], that glycolysis, specifically the Embden–Meyerhof pathway, can reverse direction in yeast. Although the trigger for flux reversal in this study was not temperature, it is pointed out that between phosphoenolpyruvate and fructose 1,6-diphosphate, many reactions stay close to equilibrium, irrespective of the flux direction. This reveals a sensitive system that is, consequently, also susceptible to thermodynamic changes.

Furthermore, many biochemical processes are based on branched reaction networks. These branch points have the potential to serve as a basis for network-inherent temperature compensation as shown in Ruoff et al. (2007) [10], but can also represent temperature-dependent control points, redirecting fluxes through different branches of a network or pathway as demonstrated here. Unfortunately, since branch points of this type would likely be embedded into bigger reaction networks, the described effects might be hidden or masked by other processes, making them hard to discriminate and identify. This work can serve as a reminder to consider the involvement of such effects when interpreting experimental results.

To design, parameterize and validate models of temperature-dependent reaction networks, it is essential to have a solid base of temperature-resolved experimental data. Although there are various experimental attempts to study the effects of temperature on different organisms, there is still a lack of data that captures the responses in sufficient scale and resolution to serve as a comprehensive foundation. Yet, looking at plants, there are measured temperature response curves for ribulose-1,5-bisphosphate carboxylase/oxygenase, better known as Rubisco, the enzyme responsible for carbon fixation in the Calvin cycle during photosynthesis [18]. These data provide a basis to begin with and there already exist models describing temperature-dependent Rubisco dynamics [19].

Here, we present a thermodynamic effect that might play a role in the accumulation of carbohydrates in plants that are exposed to cold, another plant-specific phenomenon. Although it is likely not the sole cause for this accumulation, it would be helpful to identify or rule out its contribution.

It is known that different levels of active regulation are involved and important in acclimation processes as demonstrated in Bräutigam (2009) [20], for example. Therefore, it is important to point out that the effects presented here are solely based on the structural and thermodynamic properties of the networks and require no regulation of enzyme activity on a genetic or any other level. To test the potential effect of changes in enzyme concentration and temperature, either alone or in combination, we analyzed a small toy network and demonstrated that both modifications have an impact, however, enzyme-related effects seem to be stronger than the effect of temperature changes, even though both modifiers have a non-negligible impact. We would like to emphasize that it is indispensable to incorporate thermodynamic considerations like this in the research on acclimation, as the thermodynamic influence is inevitable and its consequences might be unexpected. 

## Figures and Tables

**Figure 1 entropy-22-00117-f001:**
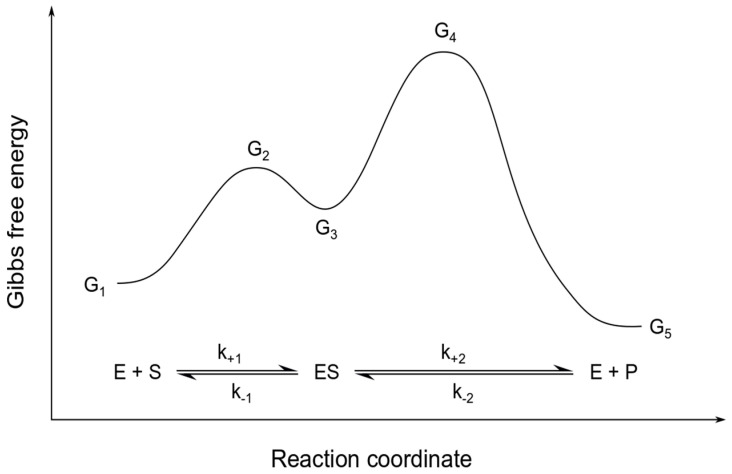
Energy diagram for reversible Michaelis-Menten kinetics. The figure sketches the different levels of Gibbs free energy *G*_1_ to *G*_5_ for a reversible enzymatic reaction with a substrate *S*, an enzyme *E* and a product *P*. The relation of these energy levels is a determining factor for the overall reaction rate and direction.

**Figure 2 entropy-22-00117-f002:**
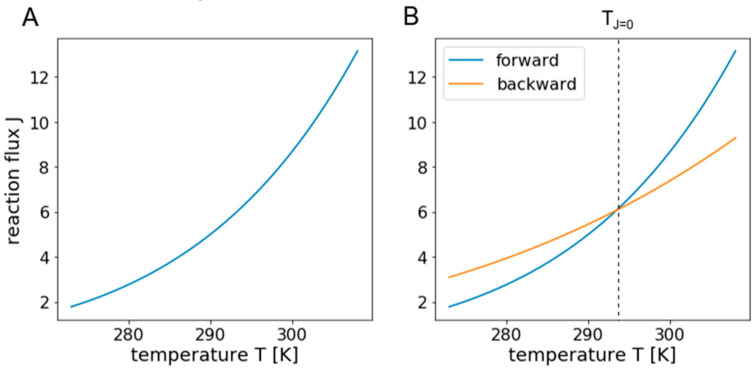
Illustration of exponential temperature response curves. Panel (**A**) shows a typical exponential temperature response curve of a reaction flux, following the Arrhenius equation. Panel (**B**) illustrates a case where the forward flux exceeds the backward flux of a reversible reaction at temperature *T_J_*_=0_. The resulting net flux at this temperature is zero and changes its direction when passing it.

**Figure 3 entropy-22-00117-f003:**
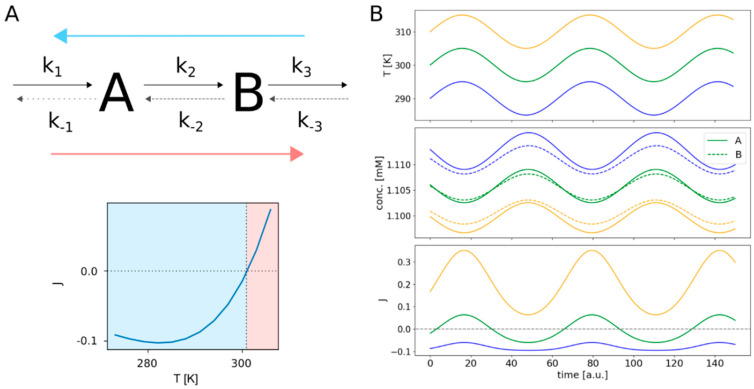
Flux reversal in a linear pathway. The upper part of (**A**) shows the structure of the reversible linear pathway, comprising the two species A and B. The rate constants of the forward reactions are indicated by *k*_1_ to *k*_3_, while *k*_−1_ to *k*_−3_ are the rate constants of the backward reactions. The lower part of (**A**) shows the corresponding net fluxes for a temperature range from 273 to 308 K. The three panels of (**B**) illustrate the system’s response to continuous cyclical temperature changes. The upper panel shows the time courses for three different temperatures oscillating by ± 5 K around 290 K (blue), 300 K (green) and 310 K (orange). The middle panel shows the resulting temporal changes in concentrations of species A and B. The flux changes are depicted in the lower panel. The colors of the concentration and flux curves match the colors of their corresponding temperature curves.

**Figure 4 entropy-22-00117-f004:**
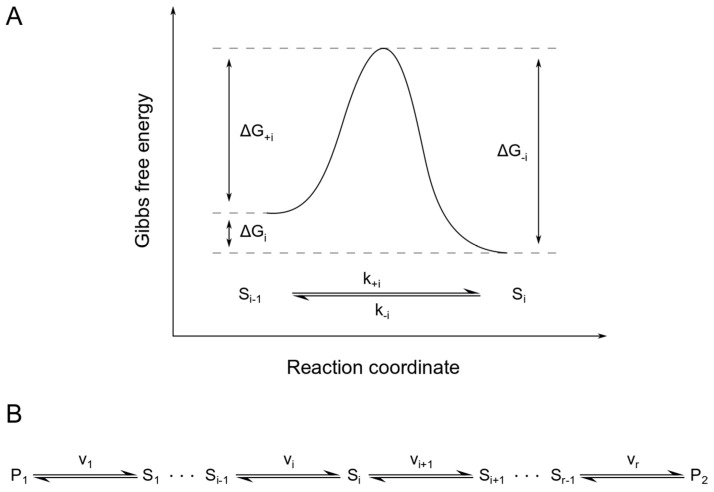
Energy diagram for a single reaction within a linear, reversible pathway. Panel (**A**) shows the Gibbs free energies of activation for the forward (*ΔG_+i_*) and backward (*ΔG_−i_*) direction of reaction i of the linear, reversible pathway shown in panel (**B**). Here, *ΔG_i_* is the difference in Gibbs free energy of the substrate S_i−1_ and the product S_i_.

**Figure 5 entropy-22-00117-f005:**
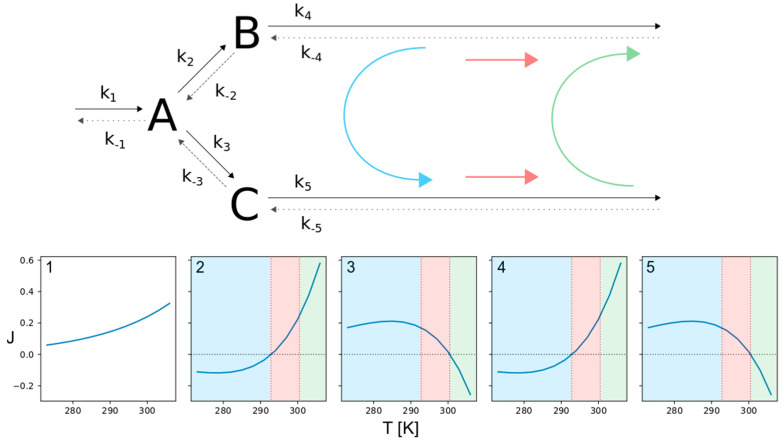
Thermo-selective, branched pathway. The figure shows the structure of a reversible branched pathway with three species (upper part) and the net fluxes (considering forward and backward direction) for all involved reactions (lower part). The numbers in top left corner in each of the panels denote the corresponding reaction. Here we see three specifiable temperature regions with distinctive flux directions (colored areas). While the flux comes in from the upper branch and flows out through the lower branch within the blue temperature region, it can change to flowing outwards at both branches (red region) and eventually invert the initial flux direction with rising temperature (green region).

**Figure 6 entropy-22-00117-f006:**
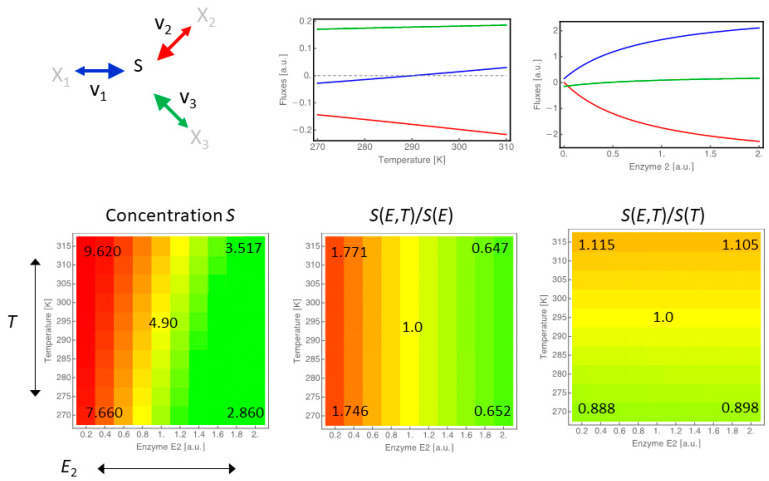
Comparison of the effect of gene expression changes versus temperature changes. Considering a minimal branched network, we changed either the concentrations of the three enzymes individually or the temperature that affects all three reactions at the same time. The panels in the top row show the flux changes in response to changes in temperature (left) and enzyme concentration (right) separately. Notice the reversal of flux direction of reaction 1 (blue). The bottom row shows three heatmaps illustrating the combined effects of enzyme 2 and temperature *T* on the concentration of species *S* (left). The relative change of *S* compared to the case that only enzyme 2 is varied and *T* is kept at 290 K (middle), and the relative change of S compared to the case that temperature is varied and enzyme 2 is kept at concentration 1.

**Figure 7 entropy-22-00117-f007:**
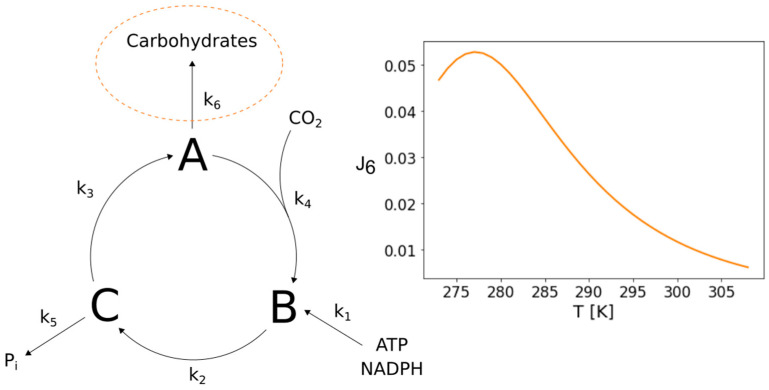
Increased flux towards carbohydrates in a minimal Calvin cycle. The figure shows the structure of a Calvin cycle model designed by P. Ruoff et al. (2007) [10] which comprises only its most essential processes as lumped single reactions. The structure of this network has the potential to exhibit an increased flux towards carbohydrates at lower temperatures that is based on the distribution of activation energies and pre-exponential factors of the reactions within the network. An accumulation of carbohydrates at lower temperatures has been observed and reported frequently.

**Figure 8 entropy-22-00117-f008:**
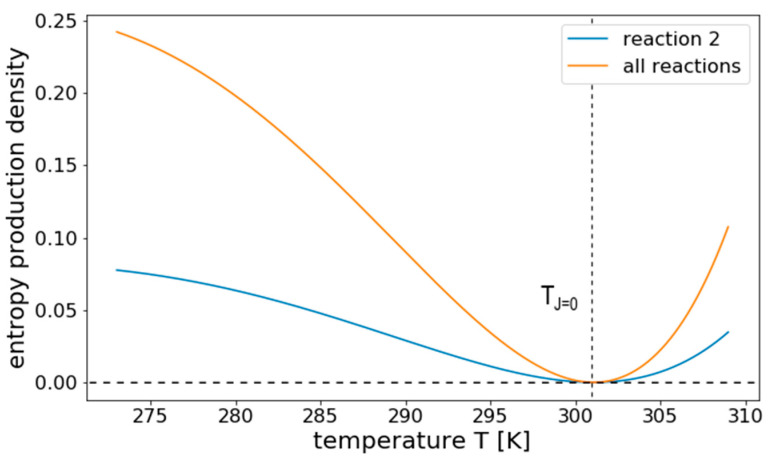
Temperature response of *σ* for a linear reversible reaction system. The figure shows the temperature response of the entropy production density *σ* of all internal reactions and reaction 2 from the linear pathway illustrated in Figure 3, separately. The entropy production vanishes at *T_J=_*_0_, the temperature at which the net flux is zero. It increases for both higher and lower temperatures.

**Table 1 entropy-22-00117-t001:** Parameter values for all four presented networks (Equations (1)–(3) and Figure 6). Indexes *i* denote the corresponding reaction with rate constant *k_i_* that is determined by *A_i_* and *ΔG_i_* following Equation (4).

*i*	*A_i_*	*ΔG_i_*
linear pathway		
1	3.00 × 10^7^	4.00 × 10^4^
−1	1.00 × 10^7^	3.75 × 10^4^
2	3.00 × 10^7^	4.00 × 10^4^
−2	1.00 × 10^7^	3.73 × 10^4^
3	3.00 × 10^7^	4.00 × 10^4^
−3	1.00 × 10^7^	3.70 × 10^4^
branched pathway		
1	2.00 × 10^7^	4.00 × 10^4^
−1	1.00 × 10^7^	4.00 × 10^4^
2	5.00 × 10^8^	5.00 × 10^4^
−2	1.50×10^7^	4.00 × 10^4^
3	1.00 × 10^7^	4.00 × 10^4^
−3	1.00 × 10^9^	5.00 × 10^4^
4	5.00 × 10^8^	5.00 × 10^4^
−4	1.50 × 10^7^	4.00 × 10^4^
5	1.00 × 10^7^	4.00 × 10^4^
−5	1.00 × 10^9^	5.00 × 10^4^
minimal Calvin cycle		
1	2.95 × 10^16^	9.20 × 10^4^
2	3.20 × 10^12^	5.00 × 10^4^
3	2.99 × 10^11^	6.00 × 10^4^
4	1.54 × 10^9^	6.00 × 10^4^
5	7.85 × 10^39^	2.10 × 10^5^
6	2.56 × 10^6^	4.00 × 10^4^

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
