# Peer review of "Chemical Reaction Networks Possess Intrinsic, Temperature-Dependent Functionality"

_entropy, 2020, doi:10.3390/e22010117_

Round 1

Reviewer 1 Report

In their article “Chemical reaction networks possess intrinsic, temperature dependent functionality”, Stephan O. Adler and Edda Klipp use a systems biology approach to simulate thermodynamic effects of temperature changes on the metabolic network in plants. Therefore, the authors simulate fluxes through selected network motives to demonstrate the functional features of the network structure that emerge temperature dependent.

The study presented by the authors is of high quality and very interesting and intriguing. In particular, it is very fascinating that the direction of fluxes can be reversed in linear pathways and selectively directed in branched systems solely by changing temperature. The work of the authors makes the excellent point that thermodynamic effects need to be considered when studying the influence of temperature change in plants. However, the impact of thermodynamic effects in comparison to other types of regulation remains unclear. Especially because the authors state themselves that changes observed in plant metabolism after temperature changes might just occur without additional regulation (L309). However, it has been shown in literature that changes of the metabolic fluxes cooccur with different types of active regulation like changes in protein concentration or enzyme activity due to posttranslational modifications. Therefore, it might be necessary to compare the impact of thermodynamic regulation with the impact of change in protein concentration.

Comments:

How does the choice of parameter influence the temperature depending switching point between the phases (Fig. 3 + 5)? It would be beneficial for the manuscript to quantify the impact of thermodynamic effects in comparison to other types of regulation (e.g. protein abundance).

Author Response

We thank the reviewer for their thoughtful review. We agree with the reviewer that cells have different ways of reacting to temperature changes and that in many cases gene expression changes are part of the response.

However, biological investigations of temperature effects currently focus on gene expression changes or other effects that are easy or at least directly measurable. Therefore, we wanted to demonstrate here that the temperature change alone can be responsible for a number of observed changes.

In order to make this even more clear, we included another small example of a branched pathway, where we illustrate the effect that either variation of enzyme concentrations or variation of temperature would have on the fluxes through the system. We also quantified the relative effect of enzyme and temperature changes. This is presented now in the new paragraph 3.3. and Figure 6. We hope that this example can convey our message.

With respect to the influence of parameter values on the effect of temperature: We also added the condition when a single reaction changes its flux direction. This is dependent on the choice of the activation energy and the pre-exponential factor. Since the kinetic parameters are also dependent on the temperature, A and Delta G are the relevant quantities here.

Lager parts of newly added text were marked as yellow and change tracking was active during editing to make all changes transparent.

Reviewer 2 Report

The manuscript titled "Chemical reaction networks possess intrinsic, temperature dependent functionality" presents work on how temperature can change biochemical reaction networks.  This work is very significant and topical, and the research is of high quality. However, I believe improvements need to be made to the manuscript before it is ready for publication.

1. Clarity of presentation of results. After reading this paper, I am still unclear on some of the major points that the paper has tried to make. I believe that some of this is due to errors in the results section. Specifically, all three panels of figure three are identical. I would encourage the authors to go over this section and clarify the results. I would also suggest that demonstrate their results with a simple simulation, e.g. A->B at one temperature, then B->A at the other. Another possible simulation to demonstrate the reversible reaction would be to have the temperature change cyclically (e.g. a sine wave) and have the chemical species transfer from A to B as a result.

2. Writing and grammar. I found many issues with the writing, which detracted from the understanding of the results. The biggest issue was that there are many places where the authors used a single sentence paragraph. This created a very disconnected and choppy reading experience. I would strongly suggest that the authors revise each of these places and integrate the single sentence paragraphs with the previous or following paragraph. They are found on lines: 53, 65, 70, 77, 108, 130, 139, 154, 157, 172, 181, 185, 187, 191, 208, 227, 281, 283, 287, 310, 320.

Minor issues.

3. Throughout the paper, the authors overuse commas in sentence. An example is the first sentence of the abstract "...live under conditions, where their ability...". The comma here is unnecessary and distracting. I suggest the authors revise this practice in whole manuscript.

4. Lines 101,103: The 'R' and 'A' constants should be italicized to match equation (4).

5. Line 107: Remove extra period

6. Table 1: Add a table caption to explain with 'i' is and give a description.

7. Subscripting in the text. Line 185, 196, both talk about "Ki" but in the equations the 'i' is subscripted. The authors should use italics and subscripting to match the parameters in the equations. This is only one example of this, they authors should fix this throughout the manuscript. Note again on line 300: "TJ", and again in the figure 7 caption.

8. Line 261: The authors have a tab at the beginning of line, it should be removed. Please check the whole manuscript for this.

9. Need to Fix references: Line 266-7: it looks like the authors made a note of which references to add here, but didn't convert them to the final numeric form. Do these citations appear in the references section?

10. Author contributions: the "formal analysis" was performed by "X.X."? This seems to have been an oversight.

Author Response

We thank the reviewer for their careful review and important hints.

We revised our manuscript according to their suggestions trying to remove all unnecessary commas, removed the one sentence paragraphs to reduce the choppiness of reading and thank the reviewer for taking the time to explicitly list them. An explanatory table caption was added and all references are in the correct numerical format now. Subscripting and italics were fixed.

We also tried to improve the clarity of presentation and language. Specifically, we added simulations with oscillating temperature to Figure 3, as suggested by the reviewer. We also removed the triplicate panel in Figure 3 (it has shown the three different fluxes, however, they are all identical in steady state).

Lager parts of newly added text were marked as yellow and change tracking was active during editing to make all changes transparent.

Reviewer 3 Report

This manuscript by Adler et al describes and models the effect of temperature on biological systems not through the indirect pathway of compensatory regulation (changes in gene expression etc) but rather through the direct effect on metabolic pathways due to differences in activation energy for different reactions in these pathways (linear or branched). I think this manuscript highlights factors which are often overlooked by biologists but I am concerned that it will struggle to find an audience in it's current form. Some of the figures could be found in very similar form in many textbooks (ex. Fig 1 and Fig 4A) and of course Figure 6 is adapted from a 2007 paper. I do not say that this manuscript is entirely unnecessary or derivative, as I mentioned, I think that this is an overlooked area. But I think that the authors need to do more to extend these findings beyond Ruoff et al, with which there is a lot of overlap, before it makes sense to publish this. As the authors point out, the best experimental system they are aware of (temperature-dependence of Rubisco) has already been extensively modelled. Perhaps by applying this same lens to a less well-studied area, they can develop a more novel manuscript.

Author Response

We thank the reviewer for their thoughtful comments. We agree that we have chosen an area where basic considerations are already textbook knowledge, but where the consequences are often overlooked by biologists studying temperature effects (by only focussing on gene expression changes).

We also agree that Figures 1 and 4A resemble textbook figures, however, we think that they are necessary to introduce the quantities that we use in our derivations.

For Figure 6 and the model from Ruoff et al.: we have chosen this model on purpose in order to use a system that is already partially understood. Ruoff et al. have focused on temperature compensation and they analyzed among others the control coefficients. Our focus is different: we would like to highlight that temperature change can directly lead to the observed flux changes towards carbohydrate accumulation when plants are exposed to cold. 

Thus, the network structure is the same, but the shown flux is from a different reaction in the network and originates from different simulations with different parameters. This should also highlight how versatile the effects of temperature changes can be already with this simple network structure.

To also provide something more novel we also added simulations with oscillating temperature to
Figure 3.

Also, we included another small example of a branched pathway, where we illustrate the effect that either variation of enzyme concentrations or variation of temperature would have on the fluxes through the system. We also quantified the relative effect of enzyme and temperature changes. This is presented now in the new paragraph 3.3. and Figure 6.

Lager parts of newly added text were marked as yellow and change tracking was active during editing to make all changes transparent.

Round 2

Reviewer 3 Report

I think the additions to this manuscript have improved the clarity and addressed some of my concerns about novelty.